



# Recommended coupling to global meteorological fields for long-term tracer simulations with WRF-GHG

David Ho[1], Michał Gałkowski[1,2], Friedemann Reum[3], Santiago Botía[1], Julia Marshall[3], Kai Uwe Totsche[4], and Christoph Gerbig[1]

[1]Max Planck Institute for Biogeochemistry, Department of Biogeochemical Signals, Jena, Germany
[2]AGH University of Kraków, Faculty of Physics and Applied Computer Science, Kraków, Poland
[3]Deutsches Zentrum für Luft- und Raumfahrt, Institut für Physik der Atmosphäre, Oberpfaffenhofen, Germany
[4]Friedrich Schiller University Jena, Institute of Geosciences, Department of Hydrogeology, Jena, Germany

**Correspondence:** David Ho (tzuho@bgc-jena.mpg.de)

**Abstract.** Atmospheric transport models are often used to simulate the distribution of greenhouse gases (GHGs). This can be in the context of forward modelling of tracer transport using surface-atmosphere fluxes, or flux estimation through inverse modelling, whereby atmospheric tracer measurements are used in combination with simulated transport. In both of these contexts, transport errors can bias the results and should therefore be minimized.

Here, we analyze transport uncertainties in the commonly-used Weather Research and Forecasting (WRF) model coupled with the greenhouse gas module (WRF-GHG), enabling passive tracer transport simulation of $CO_2$ and $CH_4$. As a mesoscale numerical weather prediction model, WRF's transport is constrained by global meteorological fields via initialization and at the lateral boundaries of the domain of interest. These global fields were generated by assimilating various meteorological data to increase the accuracy of modeled fields. However, in limited-domain models like WRF, the winds in the centre of the

domain can deviate considerably from these driving fields. As the accuracy of the wind speed and direction is critical to the prediction of tracer transport, maintaining a close link to the observations across the simulation domain is desired. On the other hand, a too close link to the global meteorological fields can degrade performance at smaller spatial scales that are better represented by the mesoscale model. In this work, we evaluated the performance of strategies for keeping WRF's meteorology compatible with meteorological observations. To avoid the complexity of assimilating meteorological observations directly,

two main strategies of coupling WRF-GHG with ERA5 meteorological reanalysis data were tested over a two-month-long simulation over the European domain: (a) restarting the model daily with fresh initial conditions from ERA5, and (b) nudging the atmospheric winds, temperatures and moisture to those of ERA5 continuously throughout the simulation period, using WRF's built-in four-dimensional data assimilation (FDDA) in grid-nudging mode.

Meteorological variables as well as simulated mole fractions of $CO_2$ and $CH_4$ were compared against observations to

assess the performance of the different strategies. We also compared planetary boundary layer height (PBLH) with radiosonde-derived estimates. Either nudging or daily restarts similarly improved the meteorology and GHG transport in our simulations, with a small advantage of using both methods in combination. However, notable differences in soil moisture were found that accumulated over the course of the simulation when not using frequent restarts. The soil moisture drift had an impact on the simulated PBLH, presumably via changing the Bowen ratio. This is partially mitigated through nudging without requiring daily



restarts, although not entirely alleviated. Soil moisture drift did not have a noticeable impact on GHG performance in our case, likely because it was dominated by other errors. However, since PBLH is critical for accurately simulating GHG transport, we recommend transport model setups that tie soil moisture to observations. Our method of frequently re-initializing simulations with meteorological reanalysis fields proved suitable for this purpose.

## 1 Introduction

Quantification of carbon sources and sinks is an area of active scientific research with implications for global warming and climate change (IPCC AR6, Rama et al., 2022). In the context of the Paris Agreement, a dramatic reduction of carbon emissions is planned in order to limit the rising of temperatures seen around the globe. Monitoring these emissions reductions is key to the success of this global effort. One method to achieve this monitoring is via atmospheric inverse modelling, wherein atmospheric transport models are used to deduce anthropogenic greenhouse gas (GHG) emissions based on atmospheric measurements of

these GHGs. Uncertainties within the inversion approach include prior emission fluxes, observation error (mainly from satellite-based measurements), and transport modeling errors (Feng et al., 2016). In the case of regional inversions, uncertainties in the lateral boundary conditions (LBCs) also need to be considered (Schuh et al., 2010). Previous studies suggest that transport errors can have considerable impact on simulated atmospheric GHG mole fractions (Lin, 2005, Díaz Isaac et al., 2014) and flux estimates (Baker et al., 2006, Lauvaux et al., 2009, Lauvaux and Davis, 2014). Given these facts, efforts must be undertaken

to reduce transport errors. As transport errors may accumulate following model initialization, this is especially important for longer simulations.

Transport models are driven by modeled meteorology fields. To reduce transport errors, one needs to use validated meteorological fields with high accuracy. The European Centre for Medium-Range Weather Forecasts (ECMWF) provides various datasets suitable for this purpose, such as reanalyses, analyses and forecasts at different spatial resolutions, up to $0.08°\times$

$0.08°$ globally. In these meteorological fields, variables such as temperature, humidity, horizontal winds and vertical mixing have a crucial impact on how accurate atmospheric transport models can be. Reanalysis data in particular, despite typically being produced at lower spatial resolution than forecasts products, are informed by historical observations and are quality controlled, which is desirable for the task.

Despite the constraint from lateral boundary conditions, if a model is initialized with a unique initial condition and is allowed

to run freely for periods of months or years, the simulated meteorology will deviate from the driving fields and therefore from reality. This is because what happens within the model domain depends on various components of the model's physics, land-surface scheme and parameterizations of sub-gridscale processes. To carry out atmospheric GHG transport simulations while maintaining a link to the reanalysis data, one of the simplest approaches is to frequently re-initialize the model with fresh meteorological initial conditions (Ahmadov et al., 2007, paragraph 25). This approach has been adopted in previous

regional studies using the Weather Research and Forecasting (WRF) mesoscale model (Skamarock et al., 2008) coupled with the greenhouse gas module (WRF-GHG, Beck et al., 2011). Specifically, regular re-initialization each day at 00:00 UTC, coupled with a 6-hour meteorological spin-up starting at 18:00 UTC the previous day has been used in Pillai et al. (2011);





Beck et al. (2013), Gałkowski et al. (2021) and others. This strategy is similar to the common practice in numerical weather prediction (NWP), where short-range forecasts are performed over limited periods due to the growth of the forecast error over

time. Such an accumulation of errors might also be expected within the regional domain (Simmons et al., 1995; Molteni et al., 1996; DelSole and Hou, 1999; Danforth et al., 2007). Another method to avoid drifting from the driving fields within a regional model is by applying nudging inside the domain (Stauffer and Seaman, 1990), ensuring that the simulated meteorological fields do not deviate too far from the global fields while the simulation is in progress. Conveniently, four-dimensional grid nudging (FDDA) is one of the built-in options within WRF (see Sect. 2.3).

Hence, some studies rely on nudging instead of frequent re-initialization of the model (Bullock et al., 2014; Spero et al., 2014; Markina et al., 2018; Zittis et al., 2018), e.g. in order to avoid discontinuity in the simulated transport (Lo et al., 2008; Vincent and Hahmann, 2015).

Few studies have assessed whether it is necessary to re-initialize periodically, or how frequently such re-initializations should take place. Others have shown that in long term continuous simulations with a focus on meteorological variables (such as winds,

temperature, pressure and precipitation), these issues can be resolved when nudging is applied (Lo et al., 2008; Vincent and Hahmann, 2015), therefore suggesting that there is no need to do multiple short runs, and continuity of the fields is ensured. However, to our knowledge, no studies have investigated the impact on long-term transport of GHG tracers, including the benefits and drawbacks of combining frequent re-initialization and nudging for long-term transport of GHGs.

The goal of this study is to determine the optimal method for keeping long-term GHG tracer simulations close to real

weather. To this end, we tested various strategies to keep WRF close to ERA5 meteorological fields, i.e. by re-initializing WRF daily, grid nudging or both. This assessment focuses on the accuracy of meteorological parameters that are critical for GHG tracer transport, namely wind speed, wind direction and planetary boundary layer height (PBLH), as well as soil moisture, surface temperature and humidity, as they impact PBLH. In addition, we simulate and evaluate transport of $CO_2$ and $CH_4$. The simulated values for both meteorological parameters and GHGs were compared to observations across the model domain. This

paper is structured as follows: Sect. 2 provides detailed information about the model setup, including the products used for meteorological and chemical initial conditions (ICs) and lateral boundary conditions (LBCs), the technique of grid nudging, the frequent restart strategy, the experimental design, and the observational data used for model validation. Sect. 3 contains the results and statistical analysis. We discuss our main findings and compare them to similar studies in Sect. 4. Finally, a summary and conclusion is given in Sect. 5.

## 85    2    Data and methods

### 2.1    WRF-GHG

The core of our system is the Weather Research and Forecasting model (WRF) version 3.9.1.1 (Skamarock et al., 2008), run with the Advanced Research WRF core (ARW). We use WRF-Chem, enabling the GHG option, which we will refer to as WRF-GHG in the text. The model uses fully compressible Eulerian non-hydrostatic equations on an Arakawa C-staggered

grid, conserving mass, momentum, entropy and scalars (Skamarock et al.; Mahadevan et al., 2008). More than a decade ago,



WRF was first used to simulate atmospheric $CO_2$ at the mesoscale (Ahmadov et al., 2007). Modules bundled with the WRF distribution, referred to as WRF-Chem (Grell et al., 2005), allow for chemical processes and the transport of atmospheric tracers to be simulated, including gases and aerosols. Such a module was prepared by Beck et al. (2011) as the greenhouse gas module (WRF-GHG), which we use in our study to simulate $CO_2$ and $CH_4$ transport. Our simulations use the Noah land-surface model (LSM) from Chen and Dudhia (2001), stochastic convective parameterization from Grell and Freitas (2014), and the MYNN2 planetary boundary layer (PBL) scheme (Nakanishi and Niino, 2006). In WRF-GHG, to simulate changes in mole fractions of $CO_2$ and $CH_4$, offline or online coupling to flux models can be used. These fluxes are associated with anthropogenic and biogenic flux components that are transported into the atmosphere to simulate the integrated signal of $CO_2$ and $CH_4$.

We compute biogenic $CO_2$ fluxes using the online VPRM model (Vegetation Photosynthesis and Respiration Model, Mahadevan et al., 2008). VPRM utilizes remote sensing products: the Enhanced Vegetation Index (EVI) and Land Surface Water Index (LSWI) derived from reflectances measured by the Moderate resolution Imaging Spectroradiometer (MODIS) satellite (http://modis.gsfc.nasa.gov/). These indices are aggregated for various vegetation types and then projected onto the model domain at the spatial resolution of the transport model. They are then combined with model-simulated solar radiation and 2-m temperature, simulating the biogenic uptake and respiration of $CO_2$ at the resolution of the model time step. Therefore, biogenic $CO_2$ fluxes differ among the experiments.

Anthropogenic $CO_2$ and $CH_4$ fluxes were taken from emission inventories. $CH_4$ emissions are from the Emission Database for Global Atmospheric Research (EDGAR) dataset, version 4.3.2 (Janssens-Maenhout et al., 2019), with a horizontal resolution of $0.1° \times 0.1°$; we have used emissions from 2012 (latest available in that product). $CO_2$ emissions were taken from the European TNO-MACC-III inventory with a spatial resolution of $0.125° \times 0.0625°$, which is an update of the earlier TNO-MACC-II dataset (Kuenen et al., 2014). Two different inventories were used for the two species because previous analysis (not shown) found that these inventories resulted in better agreement with measurements. All the emission products were re-gridded and interpolated to match the spatial resolution of the WRF domain and disaggregated from the available annual emission data into hourly emissions, using country- and sector-specific temporal and vertical profiles of GHG emissions (Brunner et al., 2019).

## 2.2 Initial and boundary conditions

For meteorological initial and lateral boundary conditions, we use ERA5 reanalysis fields (horizontal winds, pressure, temperature, sea surface temperature etc.) from ECMWF (Hersbach et al., 2020). These 3D fields were downloaded at hourly resolution on a $0.25° \times 0.25°$ regular grid (approximately 31 km spatial resolution) on the 137-level ECMWF vertical grid using the Climate Data Store Application Program Interface (https://cds.climate.copernicus.eu/, European Reanalysis 5, 2020). The data were then interpolated to WRF grids using the WRF Preprocessing System (WPS) software, which is part of the regular WRF distribution.

Initial and boundary tracer conditions for $CH_4$ and $CO_2$ were taken from the CAMS (Copernicus Atmosphere Monitoring Service) GHG short-term forecast (experiment ID: gqpe, based on IFS cycle CY43R1; see: Diamantakis and Agusti-Panareda,



2017; Agusti-Panareda et al., 2017), henceforth referred to as the CAMS fields. This product benefits from assimilation of
satellite observations at the initialization stage (from TANSO-GOSAT for $CO_2$ and $CH_4$ and also MetOp-IASI for $CH_4$).
Gałkowski et al. (2021) have shown small bias errors in this product for the free troposphere over Europe, especially for $CH_4$.
For the full period of our simulation, we have used the first 24 h of the forecast, initialized at midnight at 3-hour temporal
resolution on the ECMWF L137 vertical grid. We performed a horizontal interpolation from the original TCo1279 Gaussian

cubic octahedral grid (equivalent to approximately 9-km horizontal resolution) to an intermediate 0.125°x 0.125°regular lat-lon
grid. The 3D $CH_4$ and $CO_2$ fields were then interpolated onto the WRF grid.

## 2.3    Grid nudging

Nudging, also known as Newtonian relaxation, is a method of four-dimensional data assimilation (FDDA). This technique
keeps the model close to an analysis field over the course of a simulation. Grid nudging gently forces the model simulation

towards a series of physical reference states by adding a calculated relaxation term at every model grid point. It can be applied
selectively, i.e. above a given model level, over a selected period or throughout the simulation. This method provides a four-
dimensional analysis that is moderately balanced dynamically with driving meteorological fields and preserves continuity,
while still allowing for complex local topographical or convective variations that are resolved by the model subroutines.

These additional tendency terms are calculated by the difference of the model-state and the (re-)analysis state in the nudged

variable in addition to the normal tendency term originally derived by the model; throughout the domain or at selected altitudes,
and at each time step as shown in equation (1):

$$\frac{\partial \theta}{\partial t} = F(\theta) + G_\theta W_\theta(\hat{\theta}_0 - \theta) \tag{1}$$

where $\theta$ represents the variable fields that are nudged: two horizontal wind components ($u$ and $v$), temperature, and moisture;
while $\hat{\theta}_0$ is the value of the corresponding variable to which the nudging relaxes the solution (in this case, the interpolated

reanalysis value from ERA5). $F(\theta)$ is the tendency term obtained by the model's parameterization of physics, advection, etc.,
$G_\theta$ is a timescale constant controlling the nudging strength (nudging coefficient), and $W_\theta$ is an additional spatio-temporal
weight used to limit the effect of nudging. The nudging strength $G_\theta$ should be carefully selected, such that modeled features
are not overwritten.

For horizontal winds and temperature, $G_\theta$ was set to the default value of $3 \times 10^{-4} s^{-1}$, as several studies found it to be

acceptable (Spero et al., 2018). For moisture, a value of $4.5 \times 10^{-5} s^{-1}$ was used, following Spero et al. (2018), who found
that the default value was too high. Nudging is turned off in the PBL so that local-scale features near the surface are allowed to
develop within the regional model (Miguez-Macho et al., 2004; Lo et al., 2008; Bowden et al., 2012, 2013). Spero et al. (2014)
found that capping nudging at the tropopause (while also restricting nudging below the PBL) improved the representation of
radiation, clouds, and precipitation in WRF, which potentially affects the skill of simulating atmospheric $CO_2$, $CH_4$ and their

fluxes. (This is discussed in more detail in Sect. 4.) Nudged simulations were all started from a single pre-simulation spin-up
period (i.e., -6 hours to 00:00 UTC), in which grid nudging was applied.



Other built-in nudging methods include spectral- and observational-nudging. Moreover, grid- and observational-nudging can be used separately or in combination. For this study we focus primarily on grid nudging, but also assessed spectral nudging. However, results from spectral nudging were not significantly different and thus it is not included in our analysis.

## 2.4   Experiment design

The model domain covers Europe, spanning roughly from 30°W to 55°E and 33°N to 67°N as shown in Fig. 1. The horizontal resolution of the grid is $5 \times 5$ km$^2$ with $882 \times 705$ grid points. Simulations were performed using 60 vertical levels, with the model top at 50 hPa and 10 levels within the lowest 2 km. The internal time-step of the model was 15 s, and we used instantaneous values stored hourly in our analysis. WRF-GHG simulations were performed for May and June 2018.

Two main strategies for assuring the consistency of the WRF-GHG meteorological fields with those of ERA5 were tested. The first was to regularly (every 24 h) restart the model with fresh initial conditions from ERA5 (referred to as "DR" for "daily restarts"). The second strategy nudged the atmospheric winds, temperatures and moisture to ERA5 values continuously using the built-in FDDA (Four-Dimensional Data Assimilation) option (referred to as "GN" for "grid nudging"). In the DR simulations, the model is restarted each day at 00:00 UTC with ERA5 meteorological fields initialized at 18:00 UTC the 170   previous day, following a spin-up period of 6 hours to allow for downscaling of the variables consistent with the WRF physics. $CO_2$ and $CH_4$ tracer fields at 00:00 UTC are then copied over from the previous day's simulation (Ahmadov et al., 2012).

Six different simulations were conducted using combinations and variants of these two main strategies, as shown in Table 1. A simulation with no nudging (NN) and no restarts (NR) served as the reference simulation (combined to give NN_NR). For simulations with grid nudging, we apply nudging only above the boundary layer, so that local features below are allowed to 175   develop without interference by the coarser global analysis fields. We employed two methods for turning off nudging inside the PBL: nudging only above PBL dynamically determined by the PBL scheme (simulations denoted simply as GN) and nudging only above a certain fixed level, to avoid uncertainties in predicting PBLH (Díaz-Isaac et al., 2019). In this case we chose 3 km or 700 hPa (at the 13th model level), such that it is mostly above the boundary layer. These simulations have a suffix, resulting in GN_3km.

**Table 1.** Configuration of assimilating ERA5 for WRF simulations

| Experiment name | Configuration |
| --- | --- |
| NN_NR | Reference run - no nudging and no daily restarts |
| NN_DR | No nudging but daily restarts |
| GN_NR | Grid nudging above model-simulated PBL but no restarts |
| GN_DR | Grid nudging above model-simulated PBL and daily restarts |
| GN_3km_NR | Grid nudging fixed above 3km but no restarts |
| GN_3km_DR | Grid nudging fixed above 3km and daily restarts |





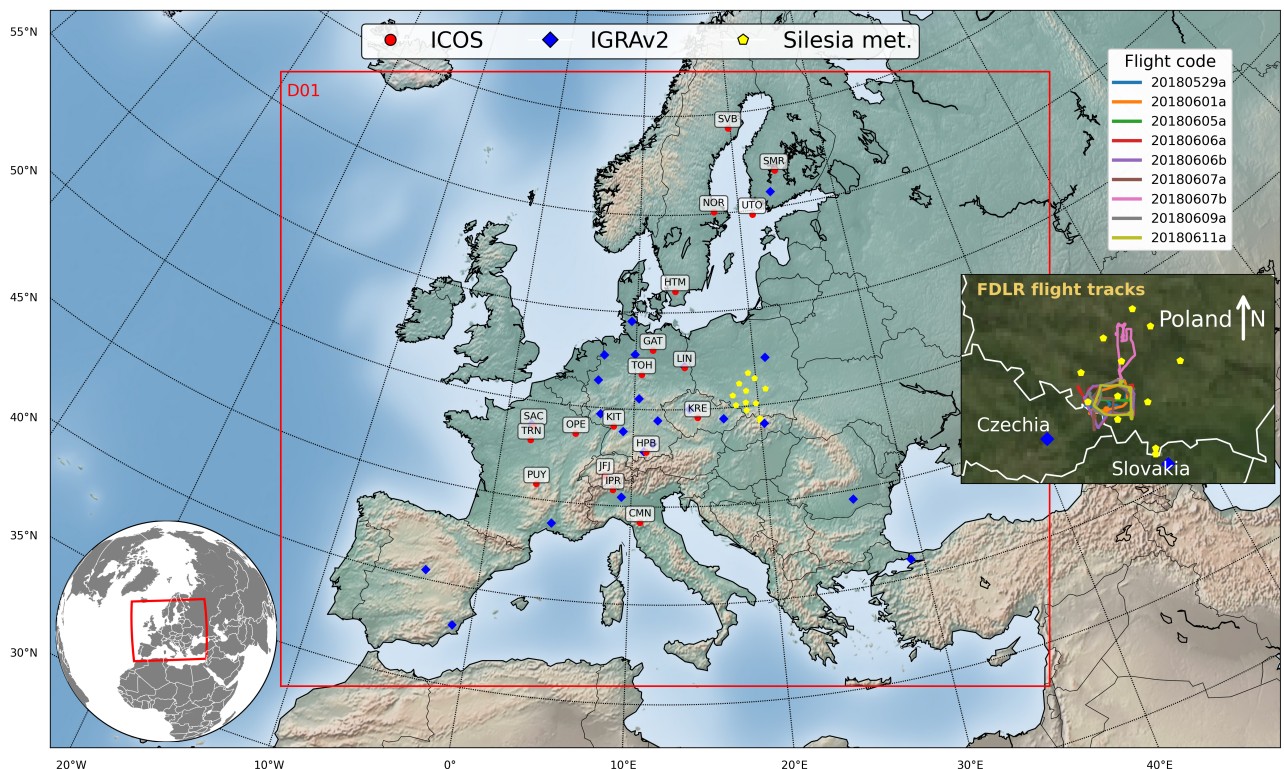

**Figure 1.** Single WRF domain over Europe with 5×5 km$^2$ grid spatial resolution. Red circles with three-letter-code labels mark the locations of ICOS tall tower sites for evaluating simulated GHG tracers mole fractions, blue diamonds show the IGRAv2 network of radiosondes for PBLH estimation, and yellow pentagons are the surface synoptic stations for assessing meteorological skills near the surface in Upper Silesia. Flight tracks of in situ $CH_4$ aircraft measurements, which were used for model-observation comparison in the Upper Silesian Coal Basin, are shown in the inset on the right.

## 2.5 Validation data and method

We compare modeled meteorological variables with observations from surface synoptic stations and radiosonde-derived PBLH estimates. Meteorological observations used for model validation are 10-m horizontal winds, 2-m temperature and specific humidity; these were taken hourly from the NOAA Integrated Surface Database (ISD, https://registry.opendata.aws/noaa-isd). A subset of the network in the Upper Silesia Coal Basin (USCB), Poland, was used, for a total of 12 stations including two mountain sites (Zakopane and Kasprowy Wierch), as shown in Fig. 1.

Additionally, we evaluated our simulations against radiosonde data from the Integrated Global Radiosonde Archive (IGRA) Version 2 (Durre et al., 2016). This network consists of quality-controlled radiosonde observations of temperature, humidity, and wind at stations across all continents. There are 22 stations of the IGRAv2 network within our model domain, as shown in Fig. 1. Only data at 12:00 UTC were used for estimating PBLH, derived using the Bulk Richardson method (Vogelezang and





Holtslag, 1996, Eq. 2) as described by Seidel et al. (2010, 2012). Thereafter, we analyze simulated soil moisture (SMOIS), i.e. the differences among simulations, and demonstrate their sensitivity to PBLH.

Apart from the meteorological evaluations, we also compared modeled GHG mixing ratios with two observation datasets. The first dataset comprised $CO_2$ and $CH_4$ measurements from 18 instrumented ICOS (the Integrated Carbon Observation System, ICOS RI et al. 2022) tall tower. We used hourly measurements from the highest intake level of each tower, collected

between 11:00 and 15:00 UTC to ensure a well-developed PBL. The second GHG dataset was low-altitude aircraft GHG measurements that were obtained in May-June 2018 during the CoMet 1.0 campaign (Fiehn et al., 2020) in the Upper Silesian Coal Basin (USCB) in southern Poland, where multiple coal mines operate. Due to the large number of mines characterized with high specific methane emissions (defined as the amount of methane emitted per 1 t of excavated coal, Swolkień et al., 2022) and continuously high levels of mining activity, this area is considered to be one of the major anthropogenic sources

of $CH_4$ in Europe, responsible for emissions of approximately 475 kt $CH_4$ / year (CoMet ED v4.01, Swolkień et al., 2022). The aircraft, equipped with an in situ analyzer for greenhouse gases, flew upwind and downwind over the mine source cluster, capturing anthropogenic signals emitted from the mines for several hours on different days during May and June 2018 under clear weather conditions. We focus the analysis on $CH_4$ as an example of a strong GHG source in the near-field.

The latter two are independent observations to evaluate the performance of each model setup, whereas ERA5 has already

assimilated data from various surface weather sites and radiosondes. Due to the availability of aircraft GHG data during this period, the comparisons against surface synoptic stations are exclusively focused on the USCB (Upper Silesian Coal Basin). The location of these measurements can also be seen in Fig. 1.

We evaluate our simulations based on how well they reproduce the observations described above. The performance metrics are mean error (ME), root mean squared error (RMSE) and coefficient of determination ($R^2$). In some cases, we show Taylor

diagrams which provide additional information.

## 3 Results

### 3.1 Meteorological validation

In order to assess meteorological model performance, we compare the results with measurements at a subset of synoptic surface stations monitoring meteorological conditions over southern Poland. We evaluate the performance of the six scenarios

mentioned in Table 1 by applying the statistical metrics on simulated and observed wind speed, wind direction, temperature and specific humidity near the model surface, namely 10-m winds (U10, V10), 2-m temperature (T2) and specific humidity (Q2); ME and RMSE were computed using data from the 12 stations in the Upper Silesian region (see locations in Fig. 1). The results are shown in Figs. 2 and 3.

With daily re-initialization introduced to the reference run (NN_NR), simulation NN_DR reduced the average RMSE for

10-m wind speed by 14.31 % (from 1.93 to 1.65 m/s, Fig. 4a), for T2 by 38.16 % (from 3.04 to 1.88 K, Fig. 4c), and for Q2 by 48.28 % (from 1.97 to 1.02 × 10⁻³ kg kg⁻¹, Fig. 4d). Grid nudging (GN_NR has) achieved slightly better error reductions with 17.61, 43.09 and 52.79 % for 10-m wind speed, T2 and Q2 in RMSE (from 1.93 to 1.59 m/s, from 3.04 to 1.73 K, and



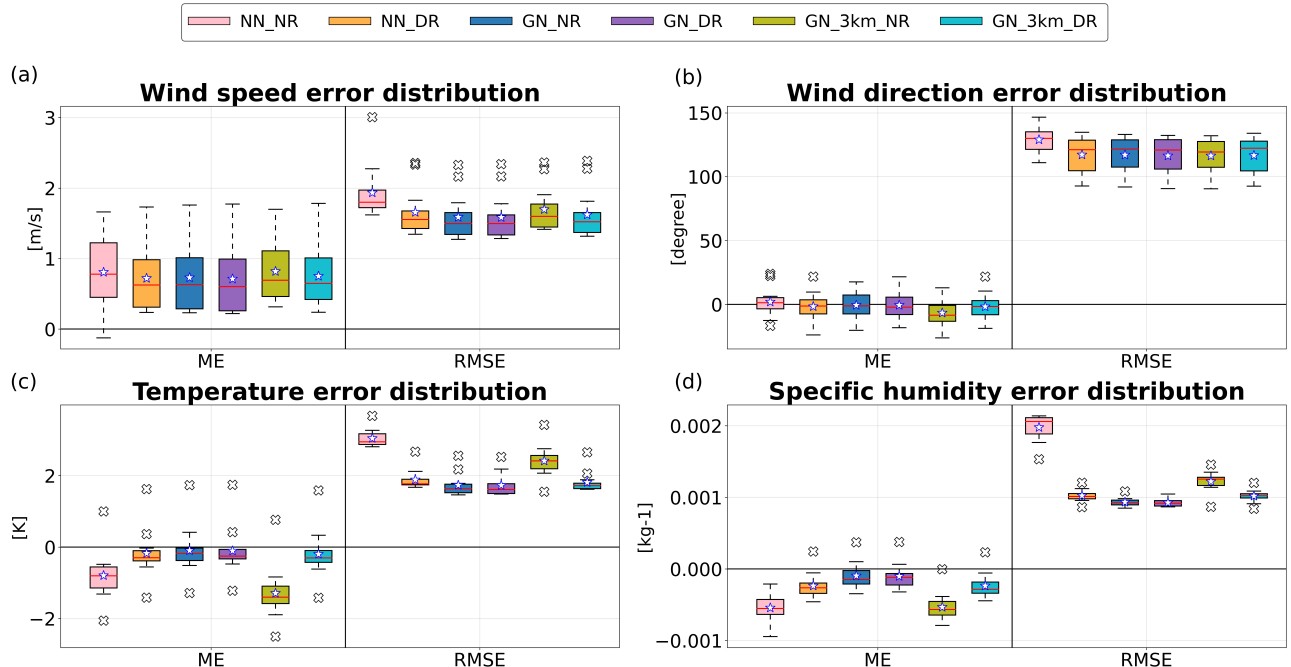

**Figure 2.** Box-whisker plots of statistical errors across 12 synoptic stations in Silesia. Analyzed hourly for May and June, 2018. The whiskers extend a distance of 1.5 times the interquartile range. The top of the box represents the 25th percentile, the bottom of the box is the 75th percentile, the red line represents median and the white star indicates the mean. Outliers are plotted with the symbol 'x'.

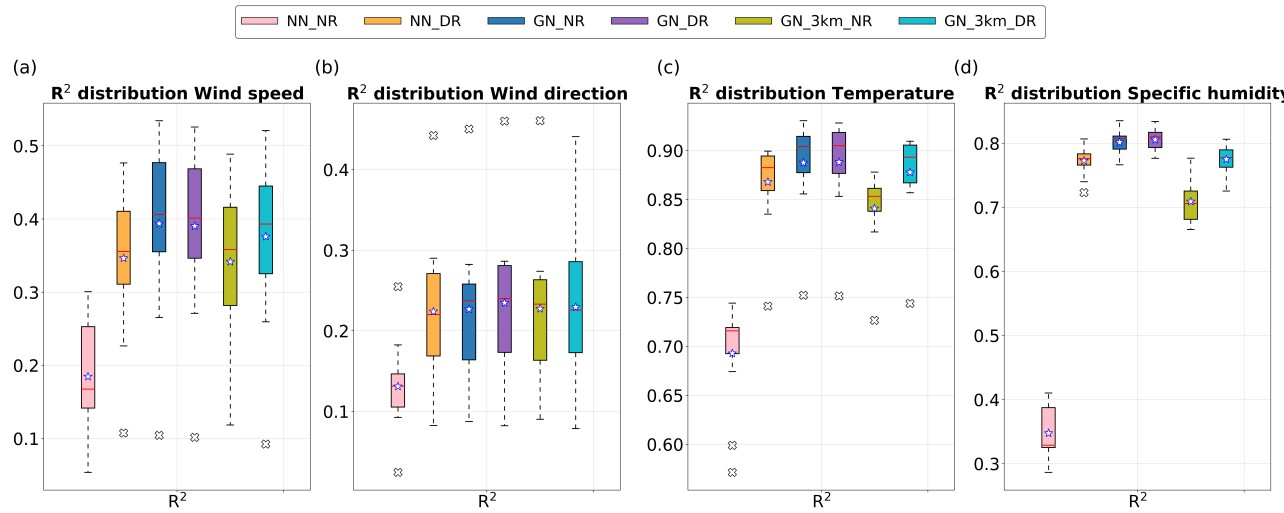

**Figure 3.** Box plots (same as in Fig. 2) of coefficient of determination $R^2$ across 12 synoptic stations in Silesia. Analyzed hourly for May and June, 2018.



from 1.97 to 0.93 × 10$^{-3}$ kg kg$^{-1}$) respectively. A similar or slightly better error reduction than GN_NR relative to the reference run (NN_NR) is achieved with simulation GN_DR, with the average RMSE of 10-m wind speed falling from 1.93 to 1.58 m/s (18.13 %), and the mean RMSE of T2 and Q2 dropping from 3.04 to 1.72 K (43.31%) and from 1.97 to 0.92 ×$10^{-3}kgkg^{-1}$ (53.28% error reduction), respectively.

Simulation GN_3km_NR, while outperforming the reference run in most metrics, has poorer performance compared to NN_DR. This is especially the case in representing T2 and Q2: compared to the reference run, the mean RMSE is only reduced to 2.40K (27.65% larger error than NN_DR) and 1.54 kg kg$^{-1}$ × $10^{-3}$ (50.7% larger error than NN_DR), respectively. A similar trend can be seen in average ME as well for 10-m wind speed, T2 and Q2. We do not observe significant variability in wind direction performance across the six different simulations. Figures 2 and 3 show the performance metrics of the four variables that are assimilated from ERA5 using FDDA. Together with statistical measures found earlier and the highest $R^2$ score, GN_DR has the best performance in general among the scenarios, with GN_NR ranking a close second.

Figure 4 shows an overview of the PBLH performance, similar to the analysis of the synoptic surface stations. The largest error reduction is again seen in simulation GN_DR, dropping 24.64 % from 618.31 to 465.95 m in RMSE compared to the reference run, closely followed by GN_NR with 21.90 % (from 618.31 to 482.4). Similar to the evaluation with surface meteorological data, GN_3km_NR also shows the smallest reduction in RMSE, and is slightly worse than NN_DR (524.93 and 502.80 m, respectively). This may be explained by the poor performance in simulating T2 and Q2, since both parameters drive the development of the PBL. All statistical results from this section are summarised in Table A1.

## 3.2 Evolution of soil moisture

We observe a divergence of SMOIS over time between scenarios with daily restarts (DR) and no restarts (NR) (Fig. 5 and Fig. 6). Soil moisture is modeled by the land surface model component of WRF (here: Noah). The key difference between the DR and NR simulations is that in the DR scenarios, soil moisture is re-initialized every 24 h from ERA5 and thus remains close to the land surface model from the ECMWF IFS (Integrated Forecast System, HTESSEL ECMWF (2016)). In contrast, for the NR scenarios, SMOIS solely follows the course of the land surface model in WRF, and, over time, drifts away from the ERA5 results (Fig. 6). In Sect. 4.3, we discuss the implications of divergent SMOIS.

## 3.3 Evaluation of WRF-GHG tracer simulations against ICOS-ATM

Figure 7 summarizes the statistical evaluation of $CO_2$ and $CH_4$ against ICOS data. In Fig. 7(a), although we observed a rather low ME for the reference scenario, we do see that the RMSE is higher than that for all other 5 scenarios. This is a sign of compensating errors of both signs. From the Taylor diagram for atmospheric $CO_2$ and $CH_4$ (Fig. 7(c)(f)), we observed a cluster separated from the reference run, in the direction of the observations (red star). This cluster consists of five simulations which all used either daily restarts and/or grid nudging. Interestingly, we do not see any significant distinction among these five scenarios. The simulations exhibit only small differences in the performance metrics for both atmospheric $CO_2$ and $CH_4$, as seen in the box plots of statistical errors and $R^2$ in Fig. 7(b)(e). By a very small margin, GN_DR yields the best performance and GN_3km_NR the worst except for the reference run. This is the same result as in the meteorological evaluation in Sect.





**Figure 4.** Statistical metrics of model performance against radiosonde-derived estimates of PBLH at 12:00 UTC for May and June, 2018. The map shows the locations of the radiosonde stations used for validation. The Taylor diagram (a) provides a summary of the skill for each simulation, showing normalized standard deviation, RMSE and correlation coefficient, averaged across all stations. Markers that lies closer to the red star (labeled as "Observed") show better performance. (b) and (c) are as in Fig. 2 and 3.

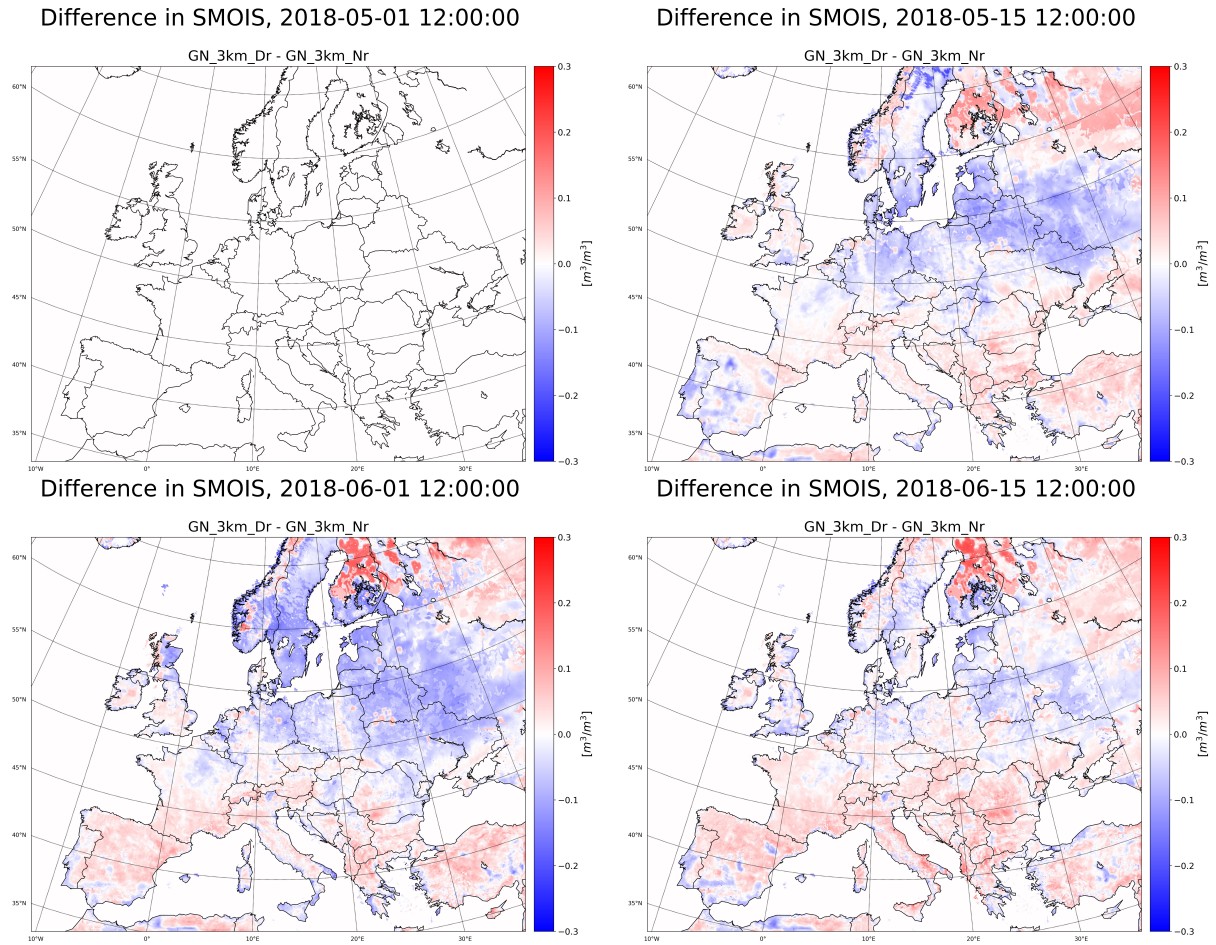

**Figure 5.** Difference in soil moisture between the simulations (GN_3km_DR - GN_3km_NR) using Grid nudging with daily restarts (GN_3km_DR) and using Grid nudging with no restarts (GN_3km_NR) above 3km.

3.1, albeit with smaller differences among all runs. For detailed statistical results for all six scenarios, readers are referred to Table A1.

ICOS tall towers are mostly situated outside of urban areas where biogenic and background signals are dominant. In order to further investigate the model performance for GHG tracers, we shift our focus to an area with a stronger influence from
anthropogenic emissions in the next section, where we evaluate the model skill using in situ aircraft measurements.

## 3.4 Evaluation against aircraft measurements

A quantitative statistical overview comparing the data from all CoMet 1.0 flights against the different model setups is shown in Fig. 8. The reference scenario (NN_NR) performed the worst among the six scenarios, and no significant performance



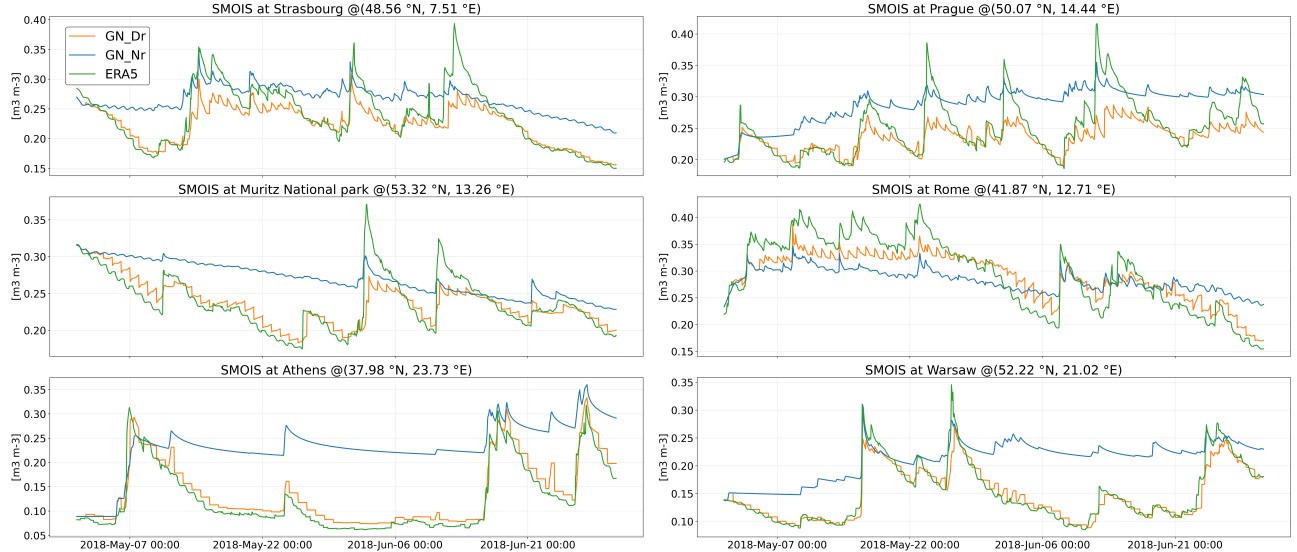

**Figure 6.** Time series showing drift in soil moisture between the simulations using Grid nudging with daily restarts (GN_DR) and using Grid nudging with no restarts (GN_NR). Within the model domain, six different locations/pixels are selected to demonstrate such discrepancies throughout the simulation period.

difference was observed among the other five setups, consistent with the comparison to tower measurements shown in Fig. 7.
A detailed overview of the time series for all flights can be found in the Supplement (Fig. S1 - S11).

Considering the spatial-temporal resolution of the model (5 km, hourly) and the assigned emissions (10 km, hourly), our simulations face limitations in capturing fine features of the plume structure when the flight path is too close to the point source in certain cases. Specifically, flights like 20180601a and 20180613a involve spiraling around point sources, with the horizontal distance from the source to the spiral loop being <5 km. As a result, our models cannot adequately represent peaks of $CH_4$
enhancements. Consequently, this led to poor statistical performance across these flights. Flights conducted outside the USCB region, such as 20180607b and 20180614a, fall under a different context for evaluating near point source emissions. Flights deemed suitable for model-data comparisons under the influence of a strong near-field source are 20180529a, 20180606a, 20180606b and 20180611a, as these flights sampled their downwind wall relatively far downstream from individual sources under a well developed PBL. (This sub-selection of flights is indicated with the asterisks in Fig. 8.)
Among these selected flights, we found an interesting case where atmospheric transport was significantly improved by grid nudging. The improvement is visible in a comparison of NN_DR and GN_DR against aircraft measurements.

On June 11 (Fig. 9), we observed a noticeable difference of approximately 40 ppb at the upwind leg and the local background between NN_DR and GN_DR, with GN_DR showing a better match with observations on that day. The underlying mechanism driving this improvement in the simulation with grid nudging in addition to daily restarts is discussed in Sect. 4.4.



**Figure 7.** Statistical overview of model performance against ICOS tall tower observations of atmospheric $CO_2$ and $CH_4$. Analysed hourly from 11:00-15:00 UTC for May and June, 2018. The first row (a-c) shows the evaluation for $CO_2$ and the second row (d-f) for $CH_4$.

## 4 Discussion

### 4.1 Performance in meteorology

The simulation GN_3km_NR performs poorly in comparison with the other simulation scenarios, based on the evaluation of skill in simulating T2 and Q2 (Sect. 3.1, Figs. 2 & 3 (c)(d)) and the $R^2$ of PBLH (Fig. 4), whereas performance differences of wind speed and wind direction are smaller. The cause of the offset in T2 and Q2 relative to observations (see ME in Fig. 2 (c)(d)) could be a consequence of the discrepancy in SMOIS, making the atmosphere wetter or dryer (Fig. 10(d)(e)). This may lead to different sensible and latent heat fluxes, which in turn could result in different temperature and humidity close to the surface. This interpretation is supported by significant dependencies of T2 and Q2 on SMOIS (Sect. 4.3.2). As there is





**Figure 8.** Quantitative statistical metrics for all flights from the CoMet 1.0 campaign for different model scenarios. Flights that crossed so close to nearby point sources that we cannot represent them well are denoted with *, flights conducted outside the USCB are denoted with **.





CH$_4$ - 20180611a

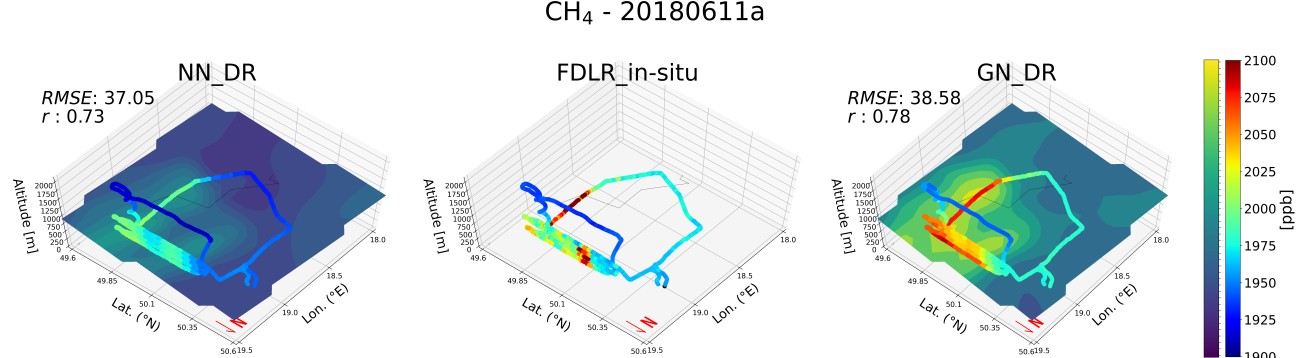

**Figure 9.** Flight tracks of the airborne platform coloured based on either modelled CH$_4$ mole fractions (left: NN_DR; right: GN_DR) or in situ aircraft measurements (middle) performed over the Upper Silesian Coal Basin. In situ observation have a high temporal resolution of 1 s spanning about 2.5 hours (ca. 9000 time steps), the models fields were stored hourly. Modeled values were extracted along the flight track from the nearest point in time and space. 2-D planes of the modeled CH$_4$ at a model level at ca. 950 m.a.s.l. are also plotted to give an indication of the plume structure and the location of the point source.

no nudging below 3 km, T2 and Q2 are not being adjusted in simulation GN_3km_NR. Simulation GN_NR also lacks daily re-initializations, just like GN_3km_NR. Nevertheless, GN_NR does not seem to show any notable biases when compared

against observation of T2 and Q2 (Fig. 2 and 3 (c)(d)). The only difference between experiments GN_3km_NR and GN_NR is the threshold altitude of grid nudging. GN_NR assimilates horizontal winds, temperature and moisture from ERA5 above the model-simulated PBL, which is dynamically diagnosed and follows its diurnal cycle. Therefore in GN_NR, each night the T and Q profiles are adjusted down to the (low) nocturnal PBL. While this is still above 2 m, T2 and Q2 are influenced by this adjustment the next day when turbulent mixing occurs. In this way, offsets in T2 and Q2 due to divergent SMOIS are

moderately amended each day by grid nudging throughout the simulation period, similar to the effect of daily re-initializations (See Fig. S25 in the Supplement). However, in Sect. 4.3.2, we demonstrate that the divergence in SMOIS, which is still present in simulation GN_NR, has a residual influence on the convective environment, if to a lesser degree than in GN_3km_NR.

    The PBLH simulation in GN_3km_NR did not perform as well as the other nudging and/or restart runs. In Sect. 4.3.2, we demonstrate that the underlying reason is likely soil moisture drift. Simulations NN_DR and GN_3km_DR have almost

identical skill in estimating PBLH. Thus, nudging only above 3 km has only a minimal impact in improving the meteorology within the boundary layer, whereas nudging dynamically above the model-simulated PBL shows better performance (seen when comparing between GN_DR and GN_3km_DR). Scenario GN_NR performs reasonably well in representing PBLH, with the second-best $R^2$ value in Fig. 4 and thus with a small advantage over NN_DR. Finally, with the addition of daily restarts, the GN_DR scenario performs slightly better than GN_NR, having the lowest ME and RMSE, as well as the highest $R^2$ among

all experiment, albeit by a small margin. In Sect. 4.3.2, we show there is a residual influence of soil moisture drift on PBLH, which likely explains the small performance advantage of GN_DR over GN_NR. In summary, either resetting the SMOIS



regime periodically using ERA5, or mitigating the impact of SMOIS drift via grid nudging, leads to a small improvement in the PBLH representation in WRF.

## 4.2 Performance in simulated greenhouse gases

In Sect. 3.3, Fig. 7(c)(f) we can see improvements relative to the reference run for both $CO_2$ and $CH_4$ when the model was re-initialized daily and/or grid-nudging was employed; however, we could not clearly distinguish the five simulations that employ either strategy from one another based on these results. This includes the GN_3km_NR simulation, which performed slightly worse than the other nudging/restart simulations in terms of PBLH. The reason for the similarity of the performances may be that other errors that are common to all simulations dominate over the impact of PBLH differences on the GHG simulation, 315 e.g. transport errors, fluxes and boundary conditions.

The evaluation with regard to the CoMet 1.0 campaign was very similar to the comparison with ICOS tower measurements, i.e. there was not much difference among the simulations. However, we uncovered a scenario that illustrates an improvement in long-range transport achieved through grid-nudging. This case is discussed in detail in Sect. 4.4. Comprehensive flight comparisons are available in Fig. S1-S23 in the Supplement.

## 320 4.3 Impact of soil moisture drift on WRF performance

Lo et al. (2008) evaluated the skill of grid-nudging with a continuous run and a weekly re-initialization run. Their results show that both simulations yield similar performance (seen also in our study (GN_NR vs. GN_DR), e.g. Fig. 2 and 3). They conclude that simulations should not be subdivided from a long simulation into shorter ones because soil parameters generally have a long memory. In the end, Lo et al. (2008) abandoned frequent restarts in favour of a continuous run but with nudging. 325 Furthermore, Vincent and Hahmann (2015) stated that the disadvantage of frequent restarts is wasted computational power for the spin-up period and discontinuities between individual simulations. However, the approach of continuous runs with nudging overlooks the impact on SMOIS. The following sections explain our findings regarding the impacts of SMOIS on model performance, specifically humidity and PBLH.

### 4.3.1 Impact of SMOIS on modeled humidity

For instances, both Bullock et al. (2014) and Zittis et al. (2018), which focus on the meteorological performance of WRF, encountered issues with surface-level water vapor being either too wet or too dry. These issues resemble what we have observed in the SMOIS differences between runs with and without restarts for different regions, as seen in Fig. 5. Therefore, SMOIS drift may explain the water-vapor discrepancies that Bullock et al. (2014) and Zittis et al. (2018) observed.

Kim et al. (2020) found that for fog simulation studies, further improvement was achieved when observed soil moisture 335 information was utilized as an initial condition. This suggests that problems in near-surface humidity can be improved by frequent restarts, informed by historical observations from quality-controlled reanalysis fields.





### 4.3.2 Impact of SMOIS on modeled PBL height

In Sect. 3.1, we showed that both nudging down to the dynamically determined PBL and daily restarts improved the simulated PBLH compared to nudging only above 3 km and not employing daily restarts. We observed the same pattern in the perfor-
340 mance of T2 and Q2. Here we show that these results can be explained by soil moisture drift in the WRF model and its effective mitigation by daily restarts or, to a slightly lesser extent, the mitigation of its impact on T2 and Q2 by grid nudging.

A positive bias in soil moisture increases humidity at the surface, and therefore the amount of energy that is being used for evapotranspiration (latent heat). This energy is then not available in the form of sensible heat, reducing temperature. The difference may be quantified by the Bowen ratio, i.e. the ratio of sensible and latent heat flux. Benjamin et al. (2016) demonstrated
that such a bias in humidity leads to a positive feedback affecting the development of the planetary boundary layer (PBL), linked with wet or dry bias. For example, large sensible heat flux triggers more turbulent convection, causing a deeper and drier PBL. The reverse is found for wetter regions. Therefore, a positive soil moisture bias results in a negative PBLH bias and vice versa, mediated by a change in the Bowen ratio. The resulting impact on the PBL is clearly not desirable for GHG tracer simulations.

To demonstrate the chain of effects that lead from soil moisture bias to PBLH bias in WRF, we compare SMOIS, T2, Q2, Bowen ratio and PBLH between simulation setups GN_DR and NN_NR, focusing on locations with radiosonde measurements at 12 UTC (Fig. 10). These simulations represent the best and worst PBLH performances, respectively, offering the most pronounced signals for our sensitivity analysis. We analyzed the difference in SMOIS against the relative difference in PBLH in Fig. 10(a), where a negative slope ($r < 0$) indicates that wetter soil leads to lower PBLH. Positive correlations ($r > 0$) between
PBLH differences and Bowen ratio discrepancies are evident in (Fig. 10(b)), suggesting that higher convection corresponds to increased PBLH. Conversely, Bowen ratio discrepancies negatively correlate with SMOIS divergence (Fig. 10(c)), implying that wetter soil is associated with reduced convection and vice versa. The influence of SMOIS on T2 and Q2 is also apparent (Fig. 10(d)(e)), confirming that the indirect effect does indeed exist in our model when the SMOIS regime is distorted, albeit the impact is subtle. (Detailed analyses for each locations can be found in the Supplement Fig. S26-S30.) These results are
consistent with the findings of Benjamin et al. (2016) and the above described mechanism how soil moisture differences have an impact on the modeled PBLH.

Consistent with the warm and dry bias reported by Benjamin et al. (2016), a bias in SMOIS will also have an effect on cloud cover and thus the shortwave radiation reaching the surface. As this is a key driver of the online VPRM module, SMOIS drift may also impact the simulated biogenic $CO_2$ fluxes (not included in this study).

### 4.4 A case study demonstrating performance improvement by grid nudging

Our analyses of the CoMet 1.0 flights revealed a case where grid nudging significantly improved the model performance. In Sect. 3.4 we mentioned the notable contrast at the upwind leg between simulations NN_DR and GN_DR in Fig. 9. In simulation GN_DR, an enhancement of $CH_4$ is seen which is regional, not close to the point source. Figure 11 shows modeled $CH_4$ over time at the location from Fig. 9 (50.26 °N, 18.47 °E). We compare between both simulations' free-tropospheric $CH_4$ at 3500





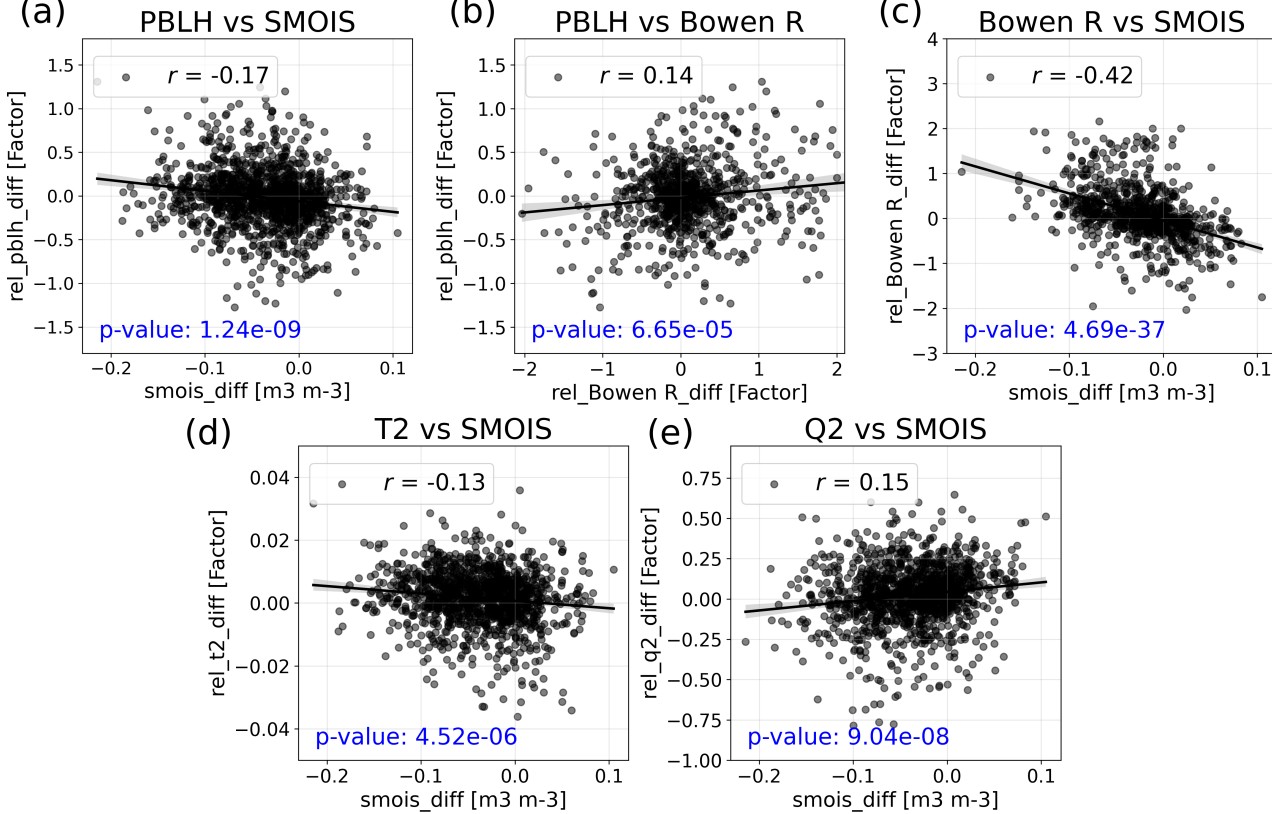

**Figure 10.** Linear dependencies among model quantities related to PBLH performance by comparing simulation GN_DR and NN_NR. Shown are the dependencies on SMOIS of the relative difference in PBLH (a), Bowen ratio (c), T2 (d) and Q2 (e). The correlation between PBLH and the Bowen ratio is shown in panel (b). The model outputs were sampled at 12 UTC at the 22 radiosonde site locations as in Fig. 4. Plots for each location can be found in the Supplement Fig. S26-S30.)

370   m.a.s.l. with observations when available. We see that the total $CH_4$ in simulation GN_DR agrees better with the observations than in NN_DR. The main contribution of the errors is not from the background, but from transport of anthropogenic $CH_4$ emitted inside the modeled domain.

We trace back in time and space to find out where the regional offset on 11 June originates and by what means. Figure 12 shows a series of snapshots of differences of simulated $CH_4$ and PBLH, which show that the creation of the regional offset

began roughly 24 hours backwards in time in northern Germany. There, the simulations show a disagreement in simulated PBLH, forming a regional enhancement of approximately 40 ppb difference in $CH_4$ within the atmosphere. This enhancement accumulated through time due to differences in PBLH, and was transported southeast to finally reach Silesia at 12 UTC on June 11, when the aircraft measurements were performed. This case demonstrates how simulated PBLH can have a critical impact on simulated GHG mixing ratios.

**Figure 11.** CH$_4$ (top: NN_DR; bottom: GN_DR) over time at Silesia (50.26 °N, 18.47 °E) in mole fraction (ppb). Background methane as CH$_4$_BCK (dashed lines) and total methane as CH$_4$_Sum, i.e. the sum of background and anthropogenic CH$_4$ (solid lines) at 3500 m.a.s.l. The white stars show coincident aircraft-based in situ measurements in the free troposphere. Note that observed values are filtered based on model-simulated PBLH to extract data in the free troposphere, hence one data point is omitted for simulation NN_DR June-13-12:00:00 UTC.



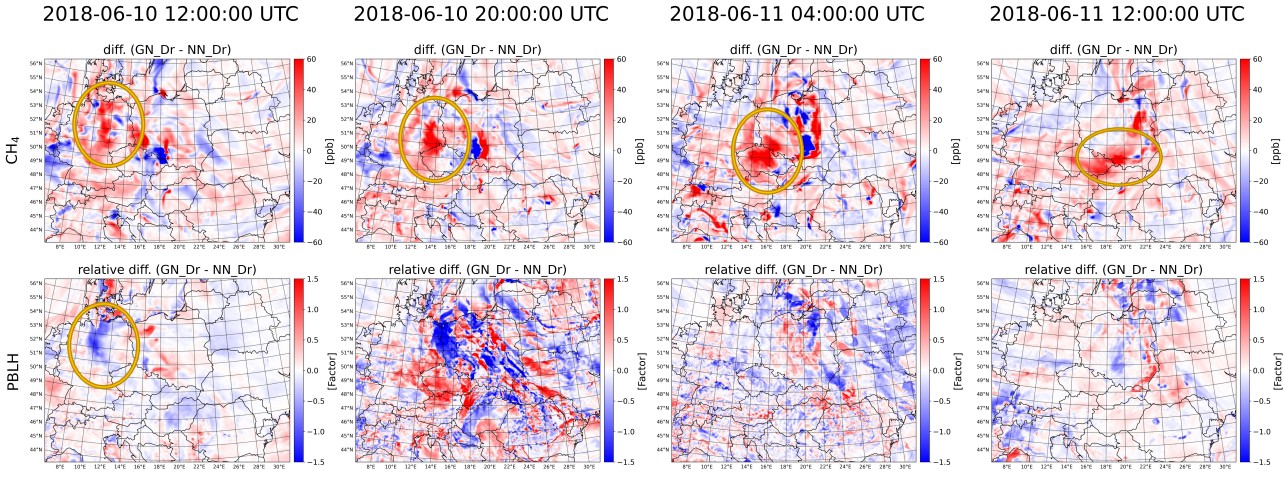

**Figure 12.** Difference in the evolution of atmospheric CH$_4$ at the 2nd model level (top) and boundary layer height (bottom), with and without grid nudging (NN_DR and GN_DR). Columns show snapshots over time, 8-hourly from 12 UTC, 10 June until 12 UTC, 11 June. The circled areas in the top row indicate the accumulated regional offset in methane being transported southeast from northern Germany to Upper Silesia. The offset originates from the difference in PBLH at 12 UTC on 10 June, also circled in the bottom row.

## 5  Summary and conclusions

Errors in atmospheric transport often limit the precision and accuracy of long-term modeling of atmospheric tracers, both forward in time, as well as in inversions for estimating GHG sources and sinks. In order to reduce this error component, we have performed a sensitivity study to determine appropriate methodologies for using ERA5 reanalyses from ECMWF to drive high-resolution (5 km horizontally) simulations of WRF-GHG over Europe. Namely, we have focused on using either the method of 1) restarting the model daily with fresh initial conditions, to maximise the consistency between WRF-simulated fields with ERA5, and/or 2) FDDA grid nudging throughout the modeled free troposphere. This applies an additional tendency term to the variables that are expected to be critical for transport (wind speed, wind direction, temperature and moisture) above a given level at each grid cell to gently force the model state closer to that of ERA5. Note that our WRF-GHG experiment does not involve nesting, unlike most previous studies. Using one large domain allowed us to assess performance across a wide range of environments, and make use of more data for model evaluation. Furthermore, in contrast to past studies that focused on meteorology, the performance differences in simulating passive atmospheric tracers was also considered.



Six different simulations with different configurations were conducted in order to assess the two main strategies outlined above, alone and in combination (see detailed description in Table 1). We found that:

1. Applying either daily restarts, nudging above the PBL or both considerably improved the performance in meteorology as well as in simulated $CO_2$ and $CH_4$ mixing ratios compared to a free run. A small advantage may be achieved by combining both daily restarts and nudging, compared to employing only one of the two methods.

2. Without re-initialization and without nudging, the selected land surface model within WRF was unable to properly represent the hydrological cycle over longer simulation periods, causing soil moisture to drift away from observation-driven reanalysis fields, which led to a deterioration of surface temperature, surface humidity and, to some extent, PBLH. Both frequent re-initializations and nudging down to the simulated PBLH alleviated the deterioration of these quantities.

3. Compared to the free-running reference simulation, grid nudging only above a fixed level of 3 km resulted in a considerably smaller performance improvement than dynamically nudging down to the model-simulated PBLH. The reason is that the latter method nudged atmospheric fields throughout nearly the entire vertical column during nighttime, when PBLHs were low, thereby improving surface temperature and surface moisture in a manner similar to the daily restarts.

4. The modeled PBLH was sensitive to soil moisture drift. The fundamental mechanism is soil moisture's influence on the Bowen ratio, i.e. an impact on the sensible heat flux that drives the development of the PBL. Nudging surface temperature and moisture minimized the impact of SMOIS drift on the PBLH performance.

5. We identified two methods that effectively alleviate the impact of soil moisture drift on modeled PBLH: restarting WRF daily and nudging down to the simulated PBL. Other methods, which we have not investigated, may be viable as well. These include employing a model that allows nudging of soil moisture or surface T2 and Q2, such as surface nudging (a built-in option within WRF), soil moisture nudging in P-X land surface model (Pleim and Gilliam, 2009), and nudging SMOIS to satellite soil moisture data (Capecchi and Brocca, 2014). However, the former two methods may run into clashes between the two different LSMs (here, WRF: Noah and ERA5: HTESSEL) during the simulation, causing inconsistencies in soil moisture dynamics. The latter requires pre-processing satellite data, and a dedicated sensitivity test for this method with our model setup. Hence, we prefer frequent restarts or nudging, as they are easier to implement.

6. Due to the impact of SMOIS drift on PBLH when nudging only above 3 km without daily restarts (simulation GN_3km_NR), and the importance of PBLH for simulating GHG mixing ratios, we expected that daily restarts or nudging down to the dynamically determined PBL would improve GHG performance compared GN_3km_NR . However, performance in GHG was very similar among those simulations. We conclude that the GHG performance in our simulations was dominated by errors other than the SMOIS drift.

7. SMOIS drift may, via its influence on precipitation, humidity, cloud formation, radiation and near-surface temperature, disturb biospheric $CO_2$ fluxes simulated by the online coupled flux model VPRM. However, dedicated sensitivity tests with a longer simulation period would be required to assess this, hence it is outside the scope of this study. Nonetheless,

we recommend addressing SMOIS drift in GHG transport model setup. This is successfully avoided by daily restarts to
constrain the land-atmosphere exchange and convective environment for longer GHG tracer simulations.

Finally, we would like to note that the frequent re-initialization approach is not only suitable for WRF-GHG: It can also be
applied to other mesoscale models simulating tracers, such as COSMO, CHIMERE, ICON-ART, etc. It ensures that models
remain consistent with the reanalysis fields, especially with respect to the land-atmosphere exchange which is associated with
convection and consequently the tracer concentration distribution.

In summary, based on this study we recommend the combination of grid nudging and frequent re-initialization of the me-
teorological reanalyses for tracer simulations over using either method alone, and consider that the additional expense of
computational time for spinups associated with daily restarts is time well spent.

*Code and data availability.* The source code of the model, WRFv3.9.1.1, is available at https://www2.mmm.ucar.edu/wrf/users/download/
get_source.html (Skamarock et al., 2008). The ERA5 dataset is freely accessible after registration from the Copernicus Climate Data Store at
https://cds.climate.copernicus.eu/ (Hersbach et al., 2020) [last access: 12/2020]. EDGAR emission inventory datasets are available at https:
//data.jrc.ec.europa.eu/dataset/jrc-edgar-edgar_v432_ghg_gridmaps (Janssens-Maenhout et al., 2019) [last access: 08/2020]. TNO-MACC-
III and CAMS data (version gqpe) can only be made available upon request. The NOAA Integrated Surface Database (ISD) was accessed on
05/2021 from https://registry.opendata.aws/noaa-isd. Radiosonde data from the IGRAv2 database are publicly available at http://doi.org/10.
7289/V5X63K0Q (Durre et al., 2016) [last access: 05/2021]. ICOS tall tower GHG measurements are available at https://doi.org/10.18160/
KCYX-HA35 (ICOS RI et al., 2022) [last access: 05/2021]. FDLR Cessna data from the CoMet 1.0 campaign are accessible from the ICOS
Carbon Portal at https://doi.org/10.18160/0SFH-JJ93 (Fiehn et al., 2020) [last access: 05/2021]. Scripts as well as processed data used for
visualisation in this paper are included on Zenodo at https://doi.org/10.5281/zenodo.10581026 (Ho, 2024). However, the WRF output data
can only be made available upon request due to the large volume (> 300 TB). Therefore, the configuration (namelists) of the WRF-GHG
simulations used for this study are also included in the same Zenodo page (Ho, 2024) to enable reproducibility.

*Video supplement.* Videos of the evolution snapshots seen in Fig. 5 and Fig. 12 are available on Zenodo at https://doi.org/10.5281/zenodo.
7347056 (Ho, 2022).

**Appendix A: Quantitative statistical metric scores**

**Appendix B: Sites used for validation**



**Table A1.** Quantitative statistical metrics for Sect.3.1 and Sect.3.3.

|  |  | NN_NR | NN_DR | GN_NR | GN_DR | GN_3km_NR | GN_3km_DR |
|---|---|---|---|---|---|---|---|
| Wind speed [$m/s$] | ME | 0.81 | 0.72 | 0.73 | 0.71 | 0.81 | 0.75 |
|  | RMSE | 1.93 | 1.66 | 1.59 | 1.58 | 1.70 | 1.62 |
|  | $R^2$ | 0.18 | 0.35 | 0.39 | 0.39 | 0.34 | 0.37 |
| Wind direction [$degree$] | ME | 1.93 | -1.69 | -0.63 | -0.71 | -6.71 | -1.77 |
|  | RMSE | 128.94 | 117.08 | 117.04 | 116.42 | 116.35 | 116.66 |
|  | $R^2$ | 0.13 | 0.22 | 0.22 | 0.23 | 0.22 | 0.23 |
| T2 [$K$] | ME | -0.79 | -0.17 | -0.10 | -0.10 | -1.29 | -0.19 |
|  | RMSE | 3.04 | 1.88 | 1.73 | 1.72 | 2.41 | 1.80 |
|  | $R^2$ | 0.69 | 0.86 | 0.89 | 0.89 | 0.84 | 0.88 |
| Q2 [$g/kg$] | ME | 5.43 | 2.36 | -3.96 | -1.04 | 5.36 | 2.37 |
|  | RMSE | 19.78 | 10.23 | 9.32 | 9.24 | 12.21 | 10.17 |
|  | $R^2$ | 0.34 | 0.77 | 0.80 | 0.81 | 0.70 | 0.77 |
| PBLH [$m$] | ME | 166.45 | 87.73 | 58.33 | 62.81 | 44.75 | 75.82 |
|  | RMSE | 618.31 | 488.00 | 488.79 | 451.04 | 550.38 | 480.85 |
|  | $R^2$ | 0.12 | 0.36 | 0.40 | 0.42 | 0.26 | 0.38 |
| $CO_2$ [ppm] | ME | 0.83 | 1.69 | 1.53 | 1.57 | 1.70 | 1.71 |
|  | RMSE | 5.74 | 4.91 | 4.93 | 4.88 | 4.98 | 4.80 |
|  | $R^2$ | 0.22 | 0.38 | 0.37 | 0.39 | 0.37 | 0.39 |
| $CH_4$ [ppb] | ME | 4.59 | 4.50 | 6.06 | 6.32 | 6.01 | 5.37 |
|  | RMSE | 27.33 | 22.86 | 22.58 | 22.50 | 22.95 | 22.84 |
|  | $R^2$ | 0.27 | 0.50 | 0.54 | 0.56 | 0.51 | 0.52 |



**Table B1.** Meteorological sites used in this study.

|  | Code/ID | Latitude, °N | Longitude, °E | Elevation, m | Name |
|---|---|---|---|---|---|
|  | 124550 99999 | 51.21 | 18.56 | 201.0 | Wieluń |
|  | 124650 99999 | 51.73 | 19.40 | 190.0 | Łódź |
|  | 124690 99999 | 51.35 | 19.86 | 189.0 | Sulejów |
|  | 125300 99999 | 50.61 | 17.96 | 163.0 | Opele |
|  | 125400 99999 | 50.05 | 18.20 | 206.0 | Racibórz |
| ISD | 125500 99999 | 50.81 | 19.10 | 295.0 | Częstochowa |
|  | 125600 99999 | 50.23 | 19.03 | 284.0 | Katowice |
|  | 125650 99999 | 50.08 | 19.80 | 236.5 | Kraków |
|  | 125700 99999 | 50.81 | 20.70 | 261.0 | Kielce-Suków |
|  | 126000 99999 | 49.80 | 19.00 | 399.0 | Bielsko-Biała |
|  | 126250 99999 | 49.30 | 19.96 | 857.0 | Zakopane |
|  | 124690 99999 | 49.23 | 19.98 | 1989.0 | Kasprowy Wierch |
|  | EZM00011520 | 50.00 | 14.44 | 302.0 | Praha-Libus |
|  | EZM00011747 | 49.45 | 17.13 | 214.8 | Prostejov |
|  | FIM00002963 | 60.81 | 23.49 | 104.0 | Jokioinen Observatory |
|  | FRM00007145 | 48.77 | 2.00 | 167.0 | Trappes |
|  | FRM00007645 | 43.85 | 4.40 | 60.0 | Nimes-Courbessac |
|  | GMM00010035 | 54.53 | 9.55 | 47.0 | Schleswig |
|  | GMM00010238 | 52.81 | 9.92 | 70.0 | Bergen |
|  | GMM00010304 | 52.71 | 7.31 | 19.0 | Meppen |
|  | GMM00010393 | 52.21 | 14.11 | 112.0 | Lindenberg |
|  | GMM00010410 | 51.40 | 6.96 | 153.0 | Essen-Bredeney |
|  | GMM00010548 | 50.56 | 10.37 | 450.0 | Meiningen |
| IGRAv2 | GMM00010618 | 49.69 | 7.32 | 376.0 | Idar-Oberstein |
|  | GMM00010739 | 48.83 | 9.20 | 314.0 | Stuttgart/Schnarrenberg |
|  | GMM00010771 | 49.42 | 11.90 | 417.0 | Kuemmersbruck |
|  | GMM00010868 | 48.24 | 11.55 | 484.0 | Muenchen-Oberschleissheim |
|  | GMM00010954 | 47.83 | 10.86 | 756.0 | Altenstadt |
|  | ITM00016080 | 45.46 | 9.28 | 104.0 | Milano Linate RDS |
|  | LOM00011952 | 49.03 | 20.31 | 703.0 | Poprad-Ganovce |
|  | PLM00012374 | 52.40 | 20.95 | 94.2 | Legionowo |
|  | ROM00015420 | 44.51 | 26.07 | 90.0 | Bucuresti Baneasa |
|  | SPM00008221 | 40.46 | -3.57 | 631.0 | Madrid/Barajas RS |
|  | SPM00008430 | 38.00 | -1.17 | 61.0 | Murcia |



**Table B2.** ICOS sites used in this study for evaluating $CH_4$ and $CO_2$.

|  | Code/ID | Latitude, °N | Longitude, °E | Elevation, m | Intake level, m | Name |
|---|---|---|---|---|---|---|
|  | CMN | 44.19 | 10.69 | 2165.0 | 8.0 | Monte Cimone |
|  | GAT | 53.06 | 11.44 | 70.0 | 341.0 | Gartow |
|  | HPB | 47.80 | 11.02 | 934.0 | 131.0 | Hohenpeißenberg |
|  | HTM | 56.09 | 13.41 | 115.0 | 150.0 | Hyltemossa |
|  | IPR | 45.81 | 8.63 | 210.0 | 5.0 | Ispra |
|  | JFJ | 46.54 | 7.98 | 3580.0 | 5.0 | Jungfraujoch |
|  | KIT | 49.09 | 8.42 | 110.0 | 200.0 | Karlsruhe |
|  | KRE | 49.57 | 15.08 | 534.0 | 250.0 | Křešín u Pacova |
| ICOS | LIN | 52.16 | 14.12 | 73.0 | 98.0 | Lindenberg |
|  | NOR | 60.08 | 17.47 | 46.0 | 100.0 | Norunda |
|  | OPE | 48.56 | 5.50 | 390.0 | 120.0 | Observatoire pérenne de l'environnement |
|  | PUY | 45.77 | 2.96 | 1465.0 | 10.0 | Puy de Dôme |
|  | SAC | 48.72 | 2.14 | 160.0 | 100.0 | Saclay |
|  | SMR | 61.84 | 24.29 | 181.0 | 125.0 | Hyytiälä |
|  | SVB | 64.25 | 19.77 | 269.0 | 150.0 | Svartberget |
|  | TOH | 51.80 | 10.53 | 801.0 | 147.0 | Torfhaus |
|  | TRN | 47.96 | 2.11 | 131.0 | 180.0 | Trainou |
|  | UTO | 59.78 | 21.36 | 8.0 | 57.0 | Utö - Baltic sea |

*Author contributions.* DH prepared the manuscript with helpful feedback from all co-authors. Model simulations and visualizations were
performed by DH with valuable technical and scientific support from all co-authors.

*Competing interests.* The authors declare that they have no conflict of interest.

*Acknowledgements.* We are grateful to the Deutsches Klimarechenzentrum (DKRZ) for providing an outstanding supercomputer platform,
Mistral & Levante (project: mj0143) where model simulations, data storage and analysis took place. In particular, we would like to thank
the MPI-BGC and the Deutscher Wetterdienst (DWD) for funding and support, under the project RiGHGorous (Research for an integrated
GreenHouse Gas monitoring system and for its use at DWD) within the extramural research programme ("Extramurale Forschung", EMF)
financed by DWD.



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
