# Peer review of "Recommended coupling to global meteorological fields for long-term tracer simulations with WRF-GHG"

_EGUsphere, 2023_

## Author Comment (AC1)

**RC1:**

The article investigates model configuration strategies to improve the accuracy of atmospheric greenhouse gas (GHG) transport simulations using the Weather Research and Forecasting (WRF) model coupled with the GHG module (WRF-GHG). The study focuses on minimizing transport errors by coupling WRF-GHG with ERA5 meteorological reanalysis data through two main strategies: daily model restarts and continuous grid nudging. Six experiments were constructed by applying different configurations of these strategies: NN_NR, NN_DR, GN_NR, GN_DR, GN_3km_NR, and GN_3km_DR through a two-month-long simulation over the European domain. The authors compared the model output with both meteorological and CO2/CH4 measurements and concluded that (1) both daily restarts and grid nudging improved meteorological accuracy and GHG transport, with a small advantage when both methods were combined; (2) notable differences in soil moisture were observed, which accumulated over the simulation period when not using frequent restarts. This drift in soil moisture affected the simulated planetary boundary layer height (PBLH) but did not significantly impact GHG performance; (3) Daily restarts or nudging minimized soil moisture drift, enhancing the simulation of surface temperature and humidity, and improving PBLH representation.

This work provides a strong, logical recommendation for the WRF-GHG setup and well-documents the results. I have been using this strategy for my simulations but haven't done or seen any work illustrating the rationale behind it. The study also provides valuable insights into optimizing long-term tracer simulations with WRF-GHG. By recommending a combination of daily restarts and grid nudging, the study offers a practical solution to enhance the accuracy of atmospheric GHG transport models, contributing to better quantification of inversion. This article has done thorough work in terms of model evaluation. The method is sound, and the results and conclusions are solid. I have no concerns regarding language or grammar. I would recommend this article after the authors address my specific comments below.

AC [response]:
*We would like to thank the referee for taking the time to read through our manuscript and for the extremely positive feedback. We appreciate the effort and are convinced that the reviewer's input has enhanced the quality of the paper.*
*We also took the opportunity to fix a few typos we noticed in the main text.*

Specific Comments:

- In the introduction, I noticed that no literature was cited beyond 2018. This raises a question about whether there has been no relevant work in this area over the past six years, which I found quite surprising. I recommend considering the inclusion of the following recent citations to provide a more comprehensive overview of the current state of research in this field:

Feng, Sha, Thomas Lauvaux, Kenneth J. Davis, Klaus Keller, Yu Zhou, Christopher Williams, Andrew E. Schuh, Junjie Liu, and Ian Baker. "Seasonal Characteristics of Model

Uncertainties From Biogenic Fluxes, Transport, and Large-Scale Boundary Inflow in Atmospheric CO2 Simulations Over North America." Journal of Geophysical Research: Atmospheres 124, no. 24 (2019): 14325–46. https://doi.org/10.1029/2019JD031165.

Feng, Sha, Thomas Lauvaux, Klaus Keller, Kenneth J. Davis, Peter Rayner, Tomohiro Oda, and Kevin R. Gurney. "A Road Map for Improving the Treatment of Uncertainties in High-Resolution Regional Carbon Flux Inverse Estimates." Geophysical Research Letters 46, no. 22 (2019): 13461–69. https://doi.org/10.1029/2019GL082987.

Gerken, Tobias, Sha Feng, Klaus Keller, Thomas Lauvaux, Joshua P. DiGangi, Yonghoon Choi, Bianca Baier, and Kenneth J. Davis. "Examining CO2 Model Observation Residuals Using ACT-America Data." Journal of Geophysical Research: Atmospheres 126, no. 18 (2021): e2020JD034481. https://doi.org/10.1029/2020JD034481.

*We thank the referee for suggesting the inclusion of these relevant publications to further enrich the introduction section of our manuscript. After reading through all three, we have now incorporated references to them in Line 35~41:*

*\*Added citation of "Feng et al., 2019a; 2019b & Chen et al. 2019" in Line 37 together with Schuh et al. 2010.*

*\* Added extra text in Line 39:*
*"The spatial resolution of the model also plays a role for emission estimates as pointed out in Feng et al., 2019b and Gerken et al. 2021. An increase in resolution may be of benefit when compared with lower resolution systems such as global models. In a study focussing on model uncertainties in regional atmospheric $CO_2$ simulations over North America, Feng et al. (2019a) found that transport uncertainties were as large as uncertainties due to biogenic fluxes in some seasons, which should be considered in the design and interpretation of inversion studies."*

- I find the justification for evaluating modeled GHG against observations to be lacking. Feng et al. (2019a; 2019b) demonstrated that flux uncertainty predominantly influences model CO2 uncertainty. This likely explains why the CO2/CH4 simulations from different experiments do not exhibit as much distinction as the meteorological variables. The authors should address this issue and provide a stronger rationale for their approach.

The authors may include these two references when they explain the causes of the similar performances in terms of GHG simulations in Line 315.

*We thank the referee for providing appropriate and relevant references to further improve our interpretation of the model evaluation against observations. Based on the two recommended references, we have added further discussion of this issue in Line 315:*
*"The interpretation is supported by Feng et al., 2019a, 2019b, who found that, despite contributions from transport and/or boundary conditions, the uncertainty in modelled atmospheric GHG mole fractions was primarily driven by the underlying fluxes, which in our case are fixed across all 6 scenarios."*

- The content and significance of Figure 8 are unclear. Regardless, I observe a consistent trend in model errors across the different flights. It appears that the reanalysis data, which provide the initial and boundary conditions for WRF-GHG, dominate the model errors. The authors should clarify the information presented in Figure 8 and discuss the impact of reanalysis data on the model's performance.

*We acknowledge that we did not address the uncertainty of the reanalysis data here. The reason is that the manuscript is mainly about how we can reduce transport errors in WRF by getting the simulation closer to ERA5 as described in the introduction section.*

*Figure 8 demonstrates that we observed a similar outcome as in Fig. 7 when comparing with aircraft measurements. As stated in the text (Sect. 3.4), the reference run (NN_NR) generally performed the worst among all scenarios, and no clear difference could be seen between the other five cases. Furthermore, Fig. 8 helps motivate the partitioning of the different flights according to their suitability for analysis. Flights outside of the USCB (marked with \*\*) tended to match the data well, but were not the focus of this study. Flights that included sampling too close to individual source regions to be represented well by simulations at 5-km resolution are marked with a single asterisk, leaving those that we think are appropriate for further analysis. All of the flights were included up to this point for transparency, to avoid creating the impression that we were cherry-picking the data used for further analysis.*

*Lastly, in the caption of Fig. 8, we added that the shown values are for $CH_4$ to clarify what is being shown, and explicitly described which subsets of flights were marked with single or double asterisks in Fig. 8 at Line 271.*

- The p-value presented in Figure 10, such as 1.24e-09, appears questionable. Could the authors clarify the number of samples used for the statistics in Figure 10? Are those r values meaningful?

[Figure]

*As mentioned in the figure caption, data were sampled at 12 UTC over a 2-month period for each radiosonde site (a total of 22 locations) when data are available, a total of 1263 samples for Fig.10 (a), (d), and (e). Notably, N/A values were excluded when calculating the Bowen ratio in several instances for Fig. 10 (b), and (c), resulting in 822 samples. For this, we improve Fig. 10 to also include the number of samples, shown at the upper-right corner as $N_\cap$ for each comparison and their data density.*

*The r-values indicate a trend that would become more prominent if SMOIS continues to diverge in time. To prevent confusion to the readers, we decided to remove the p-values from Fig. 10 , and instead inserted the following information about the significance of the relationship in the figure caption in Fig. 10:*

*"All the r values shown are significant with p-values under 0.001."*

---

## Author Comment (AC2)

**RC2:**

In this paper, the WRF model is used to simulate the transport of CO2 and CH4 for a 2-month period in 2018 at 5-km scale over Europe, with particular focus over a coal mining region of southern Poland. The authors compare the use of grid nudging and model reinitialization to improve the representation of the near-surface temperature, humidity, winds, and PBL height, and assess the implications for simulating GHGs. While somewhat similar studies have been conducted previously comparing nudging and reinitialization for simulating regional meteorology and climate, none to my knowledge have been at such high spatial resolution, nor have they focused on pollutant transport. Additionally, the linkage between soil moisture drift and PBLH errors is novel. The manuscript is well constructed and clearly written. I recommend the paper be accepted for publication after the minor issue noted below is clarified or corrected.

Lines 271-274 say that flights suitable for model-data comparisons are denoted with asterisks in Fig. 8. However, the caption to Fig. 8 says that asterisks indicate "flights that crossed so close to nearby point sources that we cannot represent them well." It appears that the authors at some point changed whether asterisks denoted "good" vs "bad" flights for comparison against the model.

AC [response]:

*We would like to thank the referee for taking the time to read through our manuscript, and for the extremely positive feedback.*

*The sentence: "This sub-selection of flights is indicated with the asterisks in Fig. 8", which was mistakenly placed after the description of the "good" flights from Line 274. This has now been moved to Line 271, directly after the description of the flights deemed inappropriate for comparison against the model. Furthermore, in response to a comment from Reviewer 1, we clarified in the text which flights were marked with a single or double asterisk, consistent with the caption of Fig. 8.*

*We also took the opportunity to fix a few typos we noticed in the main text.*

*\*Note that the line numbers here refer to the lines in the preprint version reviewed by the referees.*